# Dielectrophoresis Response of Water-in-Oil-in-Water Double Emulsion Droplets with Singular or Dual Cores

**DOI:** 10.3390/mi11121121

**Published:** 2020-12-17

**Authors:** Tianyi Jiang, Yankai Jia, Haizhen Sun, Xiaokang Deng, Dewei Tang, Yukun Ren

**Affiliations:** 1School of Mechatronics Engineering, Harbin Institute of Technology, Harbin 150001, Heilongjiang, China; jty_hit@sina.com (T.J.); jiayankaieric@gmail.com (Y.J.); shzzhen@163.com (H.S.); dkdyx485@sina.com (X.D.); dwtang@hit.edu.cn (D.T.); 2State Key Laboratory of Robotics and System, Harbin Institute of Technology, West Da-Zhi Street 92, Harbin 150001, Heilongjiang, China

**Keywords:** double emulsions, multiple droplets, dielectrophoresis, droplet manipulation

## Abstract

Microfluidic technologies have enabled generation of exquisite multiple emulsion droplets, which have been used in many fields, including single-cell assays, micro-sized chemical reactions, and material syntheses. Electrical controlling is an important technique for droplet manipulation in microfluidic systems, but the dielectrophoretic behaviors of multiple emulsion droplets in electrical fields are rarely studied. Here, we report on the dielectrophoresis response of double emulsion droplets in AC electric fields in microfluidic channel. A core-shell model is utilized for analyzing the polarization of droplet interfaces and the overall dielectrophoresis (DEP) force. The water-in-oil-in-water droplets, generated by glass capillary devices, experience negative DEP at low field frequency. At high frequency, however, the polarity of DEP is tunable by adjusting droplet shell thickness or core conductivity. Then, the behavior of droplets with two inner cores is investigated, where the droplets undergo rotation before being repelled or attracted by the strong field area. This work should benefit a wide range of applications that require manipulation of double emulsion droplets by electric fields.

## 1. Introduction

The advances of microfluidic technology have enabled the fabrication of abundant, monodisperse, and geometrically controllable double emulsion droplets, mainly including water-in-oil-in-water emulsions and oil-in-water-in-oil emulsions [1]. These droplets have been used both for scientific research, for example, drug delivery [2,3], biological culture [4,5,6], and microreactors [7,8] and for industrial products, for example, cosmetics [9,10] and food additives [11]. In these versatile applications, a library of many droplets is usually required for subsequent functioning. Sorting out or manipulation of some specific types from a droplet library is of great importance to ensure uniformity or functionality [12].

Various external fields have been implemented for droplet manipulation, including electric [13,14], magnetic [15,16], acoustic [17,18], and pneumatic methods [19,20]. Among the active sorting approaches, electric field-based droplet manipulations have drawn special attention due to their ease of control, quick response, and readiness for microscale integration. Droplets experience a dielectrophoretic force in a nonuniform electric field due to the accumulation of charges, that is, polarization at the liquid–liquid interface, resulting from the discrepancy of electric permittivity and conductivity for liquids at either side of the interface [21,22]. This phenomenon is effectively employed for manipulation of water-in-oil droplet with high efficiency [12]. However, manipulation of double emulsion droplets using electric field is rarely reported, presumably because investigation for response of double emulsion droplets under electric field is lacking. When placed in a nonuniform electric field, double emulsion droplets experience dielectrophoretic force either attractive or repulsive toward the electrodes depending on the droplet structure and the liquid property. This effect makes AC electric field a good candidate for sorting of double emulsion droplet in continuous flow. It is therefore important to investigate the dielectrophoretic properties of double emulsion droplets, which can pave the way for direct sorting or separation of multiphase droplets by electric field in microfluidic systems.

In this work, we study the dielectrophoretic response of double emulsion droplets with varying droplet geometry or with different encapsulated contents. The micro-sized droplets are prepared using a microfluidic capillary chip. Dielectrophoretic response of the droplets is investigated in a polydimethylsiloxane (PDMS) channel embedded with planar electrodes. We experimentally show that the double emulsion droplets undergo multimodal response, namely positive dielectrophoresis (DEP), negative DEP, and electrotation, in electric field. The results provide the groundwork for further droplet manipulation, including sorting and separation.

## 2. Experimental Setup and Theory

To investigate the dielectrophoretic response of double emulsion droplets, we prepare the experimental setup as depicted schematically in Figure 1. Singular or dual core double-emulsion droplets are generated by a glass capillary device for analysis. The capillary device is made by inserting two tapered circular capillaries face-to-face into a third square capillary. The outer diameter of the circular capillaries (1000 μm) is slightly smaller than the inner diameter of the square capillary (1050 μm), therefore, ensuring the coaxiality of the three capillaries. Then, a fourth thinner capillary (outer diameter 600 μm) is inserted into the left tapered capillary for generating the core of the droplet. The detailed fabrication and operation process of the capillary device is described elsewhere in our previous work [23].

For double emulsion droplets, aqueous solution is utilized as the suspending and core liquid, while the shell is composed of a mixture of PDMS oil and another silicon oil (50cSt, PMX-200, Dow Corning, Miland, MI, USA) at a ratio of 3:7. The aqueous solution used is supplemented with KCl salt, for controlling the liquid conductivity, and 2 wt% polyvinyl alcohol (PVA, 87–89% hydrolyzed, average Mw = 13,000–23,000, Sigma-Aldrich, Saint Louis, MO, USA) serving as a surfactant to stabilize the droplets. It is noted that the addition of KCl to the aqueous phase does not change the interfacial tension between the droplet shell with the suspending medium or core, which is measured to be 25 mN/m by the Wilhelmy plate method using a tensiometer.

After preparation, the droplets are collected by a syringe and reinjected into another PDMS microfluidic channel which is made by bonding a piece of PDMS slab with an electrode-patterned glass substrate. A pair of ITO (indium tin oxide) electrodes is patterned on the substrate bonding to the PDMS channel. The ITO electrode patterns are prepared by chemical etching of a ITO-deposited glass slide [24]. An AC voltage is imposed on the electrode pair to generate electric field in the channel. The droplets are placed near the electrodes to observe their response to the electric field under a microscope.

Before conducting experiments, we used a core-shell model to analyze the dielectrophoresis response of the double-emulsion droplets under electric fields [25,26]. In a nonuniform electric field, the droplets undergo inhomogeneous polarization which is equivalent to an dipole. The interaction of the AC electric field with the equivalent dipole moment gives rise to a non-zero dielectrophoretic (DEP) force [27,28] as:(1)FDEP=2πε0ε1R13RefCM∇E2
(2)$$f_{\mathrm{CM}}=\frac{{\tilde{\varepsilon }}_{23}-{\tilde{\varepsilon }}_{1}}{{\tilde{\varepsilon }}_{23}+2{\tilde{\varepsilon }}_{1}}$$
(3)$${\tilde{\varepsilon }}_{23}={\tilde{\varepsilon }}_{2}\left[{\left(\frac{R_{1}}{R_{2}}\right)}^{3}+2\left(\frac{{\tilde{\varepsilon }}_{3}-{\tilde{\varepsilon }}_{2}}{{\tilde{\varepsilon }}_{3}+2{\tilde{\varepsilon }}_{2}}\right)\right]/\left[{\left(\frac{R_{1}}{R_{2}}\right)}^{3}-\left(\frac{{\tilde{\varepsilon }}_{3}-{\tilde{\varepsilon }}_{2}}{{\tilde{\varepsilon }}_{3}+2{\tilde{\varepsilon }}_{2}}\right)\right]$$
where $\langle F_{\mathrm{DEP}}\rangle$ represents the time-average DEP force; $f_{\mathrm{CM}}$ is the Clausius–Mossotti (CM) factor; ${\tilde{\varepsilon }}_{i}=\varepsilon_{i}+\sigma_{i}/j\omega$, i = 1,2,3 is the complex permittivity of the droplet phases; $\varepsilon_{1}$, $\varepsilon_{2}$ and $\varepsilon_{3}$ are the relative permittivity of the suspending medium, droplet shell, and droplet core, respectively; $\sigma_{1}$, $\sigma_{2}$ and $\sigma_{3}$ are the conductivity of the suspending medium, droplet shell, and droplet core, respectively; and ${\tilde{\varepsilon }}_{23}$ is the equivalent complex permittivity of the droplet, as shown in Figure 2A. The factor $f_{\mathrm{CM}}$ determines the sign of the DEP force as follows: when $f_{\mathrm{CM}}>0$ the droplet experiences positive DEP force and is attracted to areas with strong electric field, for example, electrode tips; whereas if $f_{\mathrm{CM}}<0$ the droplet experiences negative DEP force and is repelled from areas with strong electric field. The sign of $f_{\mathrm{CM}}$ depends on the size ($R_{1},\;R_{2}$) and the permittivity and conductivity ($\varepsilon_{i},\;\sigma_{i}$) of the droplet. We calculate the $f_{\mathrm{CM}}$ curve as a function of frequency to better understand the DEP response of double emulsion droplet (see Table 1 for parameters). First, if keeping the outer radius of the droplet fixed, increasing the shell thickness will squeeze the CM curve toward a higher frequency and lower the frequency bandwidth of positive DEP, as shown in Figure 2B. As the shell thickness increases, the positive DEP force, which the droplet experiences, declines, and when the shell is thick enough the droplet will experience only negative DEP force. Similarly, for a given droplet size ($\delta =R_{1}-R_{2}=0.2\;\mu m$), changing the core conductivity will also mediate the DEP force for droplets, as shown in Figure 2C. The $f_{\mathrm{CM}}$ curves overlap mostly at low frequency (<1 MHz) but vary at higher frequency. As the core conductivity decreases, the positive DEP zone recedes from the left at a frequency around 1 MHz. In addition to lateral movement from DEP, the droplet also undergoes deformation and electrohydrodynamic flow due to interfacial Maxwell stress from accumulation of charges at the droplet interfaces, which has been investigated thoroughly elsewhere [29,30]. In this work, we observed that under the experimental conditions the droplet deformation or induced flow was negligible.

## 3. Results and Discussion

### 3.1. Simulation of Electric Field and Dielectrophoresis (DEP) Distribution

To obtain an in-depth understanding of the droplet behavior in electric field, we conduct a numerical simulation for DEP around the droplets by finite element analysis using a commercial software package COMSOL Multiphysics (Version 5.3) [31,32]. The same parameters are used in the simulation as in Table 1. In this numerical model, the governing equation $\nabla \left(\left(\sigma +j\omega \right)\nabla \tilde{V}\right)=0$ with $E=-\nabla \tilde{V}$ for electric field in frequency space and the time-average DEP force $\langle F_{\mathrm{DEP}}\rangle$ were calculated respectively. The electrical boundary conditions pertinent to the plane where the electrodes are deposited are summarized in Figure 3A. 

We evaluate the distribution patterns of the electric field near the electrodes and DEP force on the droplets. The dielectric layer of the PDMS-based droplet shell has intrinsic high impedance characteristics and frequency-dependent complex conductivity. Upon the signal, the circuit model for current flow path through the droplet shell can be considered to be an equivalent resistor-capacitor (RC) circuit [33]. This RC model for double emulsion consists of both the side of parallel surface conductance (R) and capacitance (C), as shown in Figure 3B. Under this equivalence, the high resistor dominates the potential drop on the shell at relatively low frequency, whereas the capacitor leads the electric current running through the shell with small potential leakage at relatively high frequency. Correspondingly, it is evident from the simulation that under electric field, at low frequency (1 kHz), the potential drop for the droplet occurs mostly on the liquid PDMS shell of the double emulsion droplet, as shown in Figure 3B. However, at high frequency (1 MHz), the electric field passes over the shell and the potential drop is distributed rather uniformly across the droplet. The same droplet has a similar distortion to the electric field near the electric field at both low and high frequencies Figure 3C. The DEP force distribution on the droplet is nonuniform, with the areas near the electrode experiencing obviously large positive and negative DEP, as shown in Figure 3D. It is also shown that the direction of DEP, indicated by arrows in the figure, can be tuned by adjusting the field frequency. The controllability of DEP direction renders the droplet behavior more versatile in AC electric fields, which is further explored with experiments.

### 3.2. DEP Dependence on Thickness

First, we study the DEP response of double-emulsion droplets with fixed thickness [33], as shown in Figure 4A,B. A water-in-oil-in-water double emulsion droplet with diameter of 340 μm and shell thickness of 1 μm is placed near an ITO electrode. The conductivities of the suspending medium and the droplet core, adjusted by adding KCL, are controlled to be 8 and 1.5 S/m, respectively. Upon application of an AC electric field with frequency of 50 kHz, the droplet is repelled from the strong field area, i.e., electrode edge, because of negative DEP. The DEP force exerted on the droplet, according to the simulation in Figure 3, decreases as the droplet moves away from the electrode edge. Therefore, the speed of droplet movement also decreases. Finally, the droplet settles down and keeps stationary when the DEP force is not large enough to overcome the viscous force. The double-emulsion system has two relaxation frequencies, one for each of the two interfaces, i.e., the suspending medium-shell interface and the shell-core interfaces. At low frequency (e.g., 50 kHz), polarization of the droplet occurs primarily at the outer interface. The whole droplet can be seen as if it consisted of only the shell material. And the polarization of the medium-shell interface is much greater outside the droplet than that inside, rendering a negative DEP on the droplet which agrees with the experimental result in Figure 4A.

At frequencies above a certain value, i.e., the cutoff frequency, the polarization of a droplet shifts to the core-shell interface, resulting in a positive DEP experience by the droplet since the core with higher conductivity is polarized greater than the medium. To verify this, the dependence of droplet DEP on field frequency is investigated by increasing the field frequency to 2 MHz, as shown in Figure 4B. At 2 MHz, the same droplet indeed experiences positive DEP and is dragged toward the electrode. The speed of droplet movement increases as it gets closer to the electrode edge, resulting from an increasing DEP force closer to the strong field area. The cutoff frequency is tested to be about 1 MHz, which is in good agreement with the calculation shown in Figure 2B and with the simulation in Figure 3D.

Adjusting the shell thickness can change the characteristic frequency for DEP of the droplets. By tuning the flow rates when generating droplets [34], the thickness of droplet shell is reduced to 200 nm, as shown in Figure 4C. According to calculation in Figure 2B, the DEP cutoff frequency for 200 nm droplet is 700 kHz. Droplets with different cutoff frequencies behave differently in the same electric field at an intermediate frequency range. To test this, two types of droplets with shell thickness of 1 μm and 200 nm are placed in the microchannel. When subjected to an AC electric field with frequency of 1.5 MHz, the droplet with thinner shell (200 nm) experiences positive DEP force and is attracted toward the electrode edge. However, the droplets with thicker shells (1 μm) are repelled and move slightly away from the electrode edge, indicating that they are experiencing negative DEP force. This agrees with the plot in Figure 2B, which shows that the $f_{\mathrm{CM}}$ curves shift towards the left (lower frequency) as the shell thickness decreases.

### 3.3. DEP Dependence on Core Conductivity

In addition to droplet size, there are circumstances where the encapsulated content of double emulsion droplets varies from case to case, consequently, their electrical responses are also different. Then, we investigate the influence of core conductivity on droplet response to AC electric fields, as shown in Figure 5. At low core conductivity (15 mS/m) and frequency of 3 MHz, as shown in Figure 5A, the droplet is attracted towards the electrode edge because of a positive DEP force. The time scale in the sequential snapshots shows how fast the droplet reacts to the applied electric field and also how strong the DEP force is that is exerted on the droplet. The long time (6.2 s) the movement takes indicates a weak DEP force for the droplet. However, when the core conductivity increases to 1.5 S/m and the field frequency is still 3 MHz, as shown in Figure 5B, the droplet is dragged toward the electrode with shorter time for a longer distance. This shows that the droplet experiences a much larger DEP force with higher core conductivity. This agrees with the calculation in Figure 2C that for higher core conductivity the positive DEP force is larger, and the force dominates at a broader frequency width. Under nonuniform electric field, larger ion strength leads to a larger group of ions accumulating at the interface, which resulting in larger DEP force.

### 3.4. DEP for Dual-Core Droplets

One prominent feature for double emulsion systems is the ability for the droplets to accommodate more than one core with different contents [35]. Then, we study the behavior of double emulsion droplets with two cores, as shown in Figure 6. When placed in an electric field, the dual-core droplet is polarized because of the discrepancy of dielectric constant and electric conductivity between the droplet and the suspending medium. The distribution of electric field in the PDMS layer is highly nonuniform, with the field maximized between the two inner cores and at the thin area of the PDMS layer. The droplet is more polarized at the area with high electric field. After polarization, the whole droplet can be considered to be a dipole moment of which the direction is not parallel to the electric field. The action of the electric field to the dipole moment results in a torque applied on the droplet; therefore, the droplet rotates accordingly [34]. When subject to a electric field of 1.5 MHz, the droplet rotates at first, which is originated from the alignment of the induced dipole moment with the electric field [24]. After rotation, the droplet is repelled away from the electrode edge when experiencing negative DEP with low core conductivity (15 mS/m), as shown in Figure 6A; whereas, when experiencing positive DEP with high core conductivity (1.5 S/m), the droplet is dragged toward the electrode edge, as shown in Figure 6B. 

## 4. Conclusions

In summary, we have investigated the dielectrophoretic response of double emulsion droplets in AC electric field. The core-shell model is readily used for analyzing the DEP response of double emulsion droplets. The DEP response can be tuned by droplet shell thickness or core conductivity. At low field frequency, the droplets experience negative DEP. At high field frequency, the droplets experience positive DEP if the shell thickness is small (e.g., 200 nm), whereas it switches to negative if the shell is thick enough (e.g., 1 μm). Additionally, core conductivity affects the magnitude and frequency width of the positive DEP experienced by the droplets. A higher core conductivity increases the magnitude and frequency width for positive DEP. Moreover, dual core droplets rotate first when subject to electric fields, after which they will be repelled or attracted by the strong field area according to their properties.

The motivation of this work is to provide a controllable way to manipulate double emulsion droplets in microfluidic channels. By electrical repulsion or attraction droplets from the electrode, their movement can be controlled flexibly. In a typical scenario, the medium suspended with double-emulsion droplet flows continuously through a microchannel, applying an external DEP force can be used for sorting or separating targeted droplets from bulk droplets, as is used for single emulsion droplets. A limiting factor of this method is that the discrepancy of electrical parameters between droplets needs to be large enough for effective separation. This can be solved by combining this technique with other manipulation method, for example, fluorescence detection [35]. This work has the potential to be used for further manipulation of double emulsion droplets, including sorting [12] and separation [36].

## Figures and Tables

**Figure 1 micromachines-11-01121-f001:**
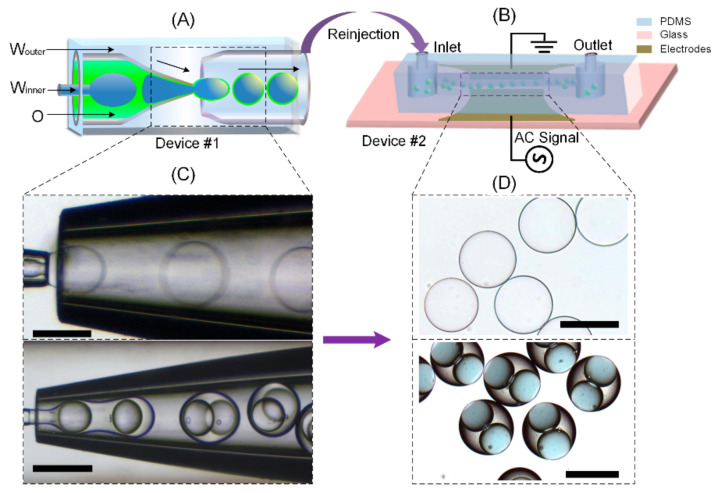
Generation of double emulsion droplets with single and dual cores. W_outer_, outer aqueous phase; W_inner_, inner aqueous phase; O, PDMS oil phase. (**A**) Schematic illustration of the glass capillary chip for droplet preparation; (**B**) The PDMS chip for droplet manipulation. The microscopic snapshots of droplet generation in the microfluidic chips and the droplets after generation are shown in (**C**) and (**D**), respectively. Scar bar, 300 μm.

**Figure 2 micromachines-11-01121-f002:**
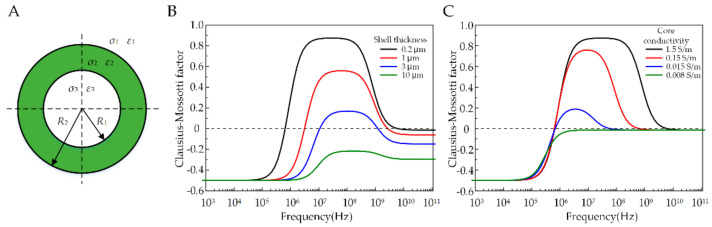
Calculation of Clsusius–Mossotti factor as a function of field frequencies. (**A**) The core-shell model for double emulsions; (**B**) CM factor plot with varying shell thickness, $\sigma_{3}$ = 1.5 and $\sigma_{1}$ = 0.008; (**C**) CM factor plot with varying core conductivity, $\sigma_{1}$ = 0.008 and δ = 200 nm.

**Figure 3 micromachines-11-01121-f003:**
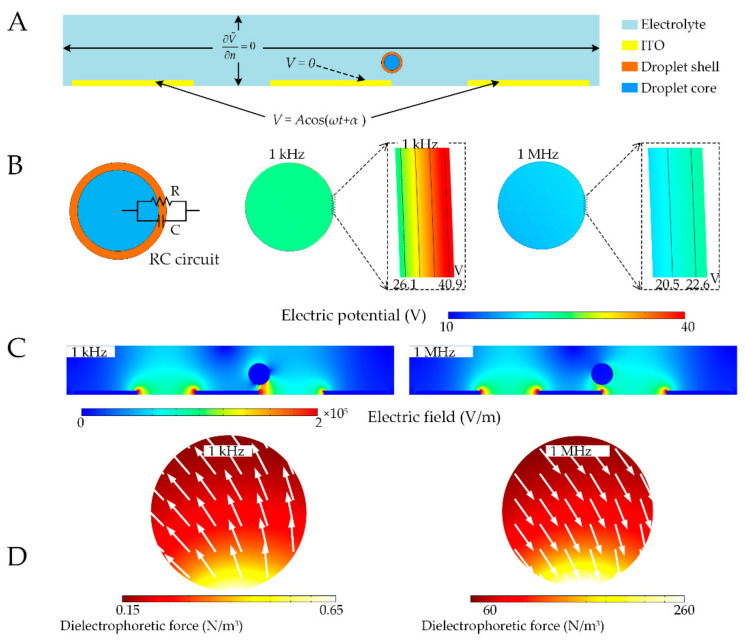
Simulation of dielectrophoresis (DEP0 response of double emulsion droplets. (**A**) The established geometrical model for numerical simulation; (**B**) Circuit model for current flow path through a droplet shell and the corresponding voltage drop across this shell at a voltage of 100 V; (**C**) Electric field distribution near the electrodes; (**D**) Dielectrophoretic force imposed on the droplets.

**Figure 4 micromachines-11-01121-f004:**
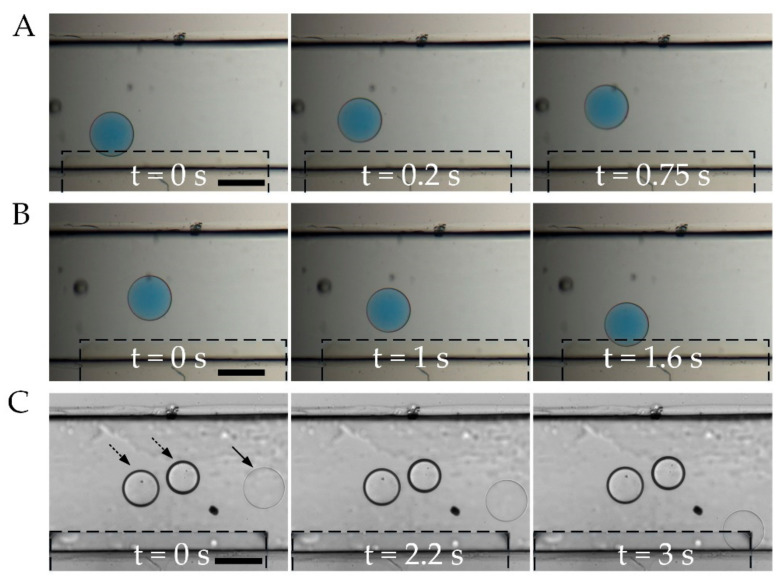
DEP response of droplet with different shell thicknesses (Appendix A). (**A**) Negative DEP response of droplet (shell thickness 1 μm) at the field frequency of 50 kHz; (**B**) Positive DEP response of droplet (shell thickness 1 μm) at the field frequency of 2 MHz; (**C**) Difference of droplet response to the electric field with different shell thickness. Scale bar, 200 μm.

**Figure 5 micromachines-11-01121-f005:**
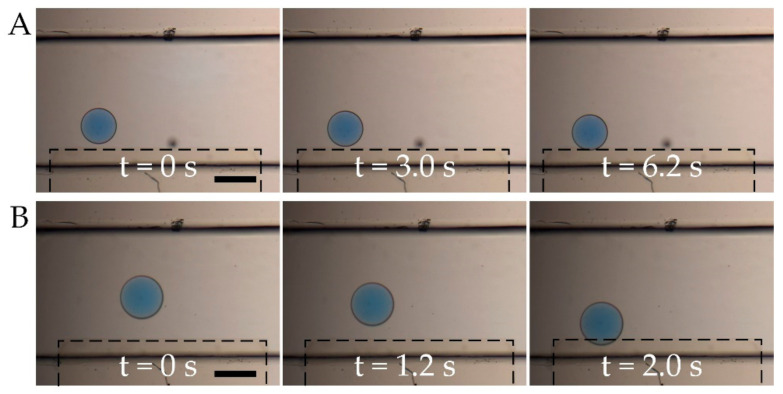
Influence of droplet DEP response on droplet core conductivity (Appendix A). (**A**) Low core conductivity, 15 mS/m; (**B**) High core conductivity, 1.5 S/m. Scale bar, 300 μm.

**Figure 6 micromachines-11-01121-f006:**
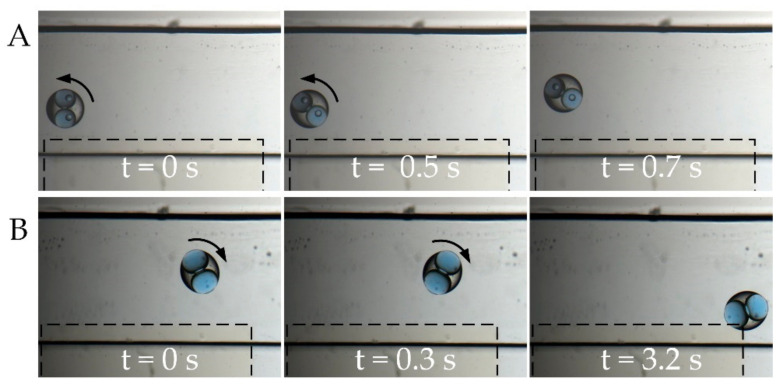
DEP response of droplets with two cores (Appendix A). (**A**) Negative DEP response with low core conductivity, 15 mS/m; (**B**) Positive DEP response with high core conductivity, 1.5 S/m. Scale bar, 300 μm.

**Table 1 micromachines-11-01121-t001:** Parameters and dimensions used in the modeling calculation.

Symbol	Implication	Value
$\varepsilon_{0}$	Permittivity of free space	8.854 × 10^−^^12^ F/m
$\varepsilon_{1}$	Relative permittivity of suspending medium	80
$\varepsilon_{2}$	Relative permittivity of droplet shell	2.3
$\varepsilon_{3}$	Relative permittivity of droplet core	80
$\sigma_{1}$	Conductivity of suspending medium	8 × 10^−^^3^ S/m
$\sigma_{2}$	Conductivity of droplet shell	2.5 × 10^−^^12^ S/m
$\sigma_{3}$	Conductivity of droplet core	0.015, 0.15, 1.5 S/m
$R_{1}$	Droplet outer radius	170 μm
$R_{2}$	Droplet inner radius	169.8, 169, 167 μm
δ	Droplet shell thickness	0.2, 1, 3 μm
L	Electrode length	1 mm
$L_{s}$	Electrode distance	1 mm
M	Width of microchannel	650 μm

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
