# Peer review of "Dielectrophoresis Response of Water-in-Oil-in-Water Double Emulsion Droplets with Singular or Dual Cores"

_micromachines, 2020, doi:10.3390/mi11121121_

Round 1

Reviewer 1 Report

This paper investigates the dielectrophoretic response of water-in-oil-in-water double emulsion droplets with single or dual cores. In the manuscript, the DEP response for dual core droplet is not so mentioned. What causes the rotation of dual core droplet? I think authors should discuss the reason or references.

I commented for the manuscript as follows.

L12

Sing-cell

L55

Undergoe

Figure 1(a)

Is “O” oil?

Figure 1(c),(d)

I think authors should use the same scale for capillaries and droplets, respectively.

L79

“which is measured to be 25 mN/m”

By what?

LL95-98

These sentences correspond to Fig.2(a), so please add the comment for fig. 2(a).

L107,110

I think figure 2 is referred.

In addition, a,b,c are used in Fig. 1 and A,B,C are used for Fig. 2-6. Authors should use consistent form.

L127

How is the applied voltage?

Figure 3A

Where is the boundary of liquid PDMS shell?

Figure 3C

Please add frequencies as A and B.

Section 3.4

What causes the rotation of dual core droplet? I think authors should discuss the reason or references.

Whole

There are some errors.

Ex.) Mhz, KHz, undergoe,….

English revision is necessary.

Reviewer 2 Report

Dielectrophoresis Response of Water-in-oil-in-water 3 Double Emulsion Droplets with Singular or Dual 4 Cores

The paper discusses the effect of dielectrophoretic force with the change in thickness of the droplet shell, conductivity and presence of droplet in droplet in case of droplet microfluidics. The material is presented with clarity; however this manuscript does not present considerable novel scientific results and thus could be published as a technical report rather than a manuscript.

  1. The authors have not provided a strong motivation with suitable applications to support why this study is important. Droplet manipulation using dielectrophoresis has been done before, thus it is important to stress on what new scientific contribution is made by this manuscriptand why it would be important for the field.
  2. Authors need to justify why they chose the shell thickness which have been specified in their work. The 1 µm and 200 nm shell thicknesses have been used here, but these values have no basis. Similarly, provide justifications for choosing the droplet size and conductivities used here.
  3. It will be useful to the reader to understand the type of simulation model and equations that are solved by the model. Please provide these details.
  4. The computational results do not provide any new knowledge in the present form. Try to relate the information that is obtained from the computational results with the experimental results.
  5. Please explain the results on lines 128-135 in more detail and discuss the phenomena that is occurring.
  6. The experimental results are qualitative and do not provide any new information. The authors need to quantify these results to understand how the thickness of the shell, the distance from the electrode and the position of the droplet with respect to the electrode can affect the DEP force.
  7. Justify the importance of using the time scale for the images provided in figure 5.
  8. The results pertaining to electro-rotation need to be provided in greater detail. Reasoning for the phenomena that is occurring and the effect of size of droplet, double emulsions on this need to be discussed in the manuscript.
  9. Manuscript needs to have a good discussion section to interpret the results and state their relevance to the real-world applications. providing a comparison, the some of the published works in this field will also be useful.
  10. Please proofread the manuscript for spelling, grammar and punctuation errors. Please place the punctuation marks after the reference. For eg. [2], instead of ,[2].

Round 2

Reviewer 1 Report

The manuscript is improved. I think this paper is acceptable for publication.

Author Response

We thank the reviewer for the recommendation for publication in micromachines.

Reviewer 2 Report

Authors have done a fair job answering the questions. There are a few things which need to be made clear.

  1. Provide a short paragraph/reference for your COMSOL model. Did you use a user defined model or a pre-existing one? What equations are you solving? What were the boundary conditions?
  2. Section 3.1: The discussion of the potential drop with resistor capacitor model needs better explanation. The section has few errors which is making it difficult to understand. Please simplify the explanation and provide references for the behavior seen here. It might be helpful to plot V/m in Figure 3B instead of V.
  3. Line 162: It is difficult to understand what this sentence means. Please simplify.
  4. If you can relate your experimental results from section 3.2 and 3.3 with simulations or explain the behavior with simulations, it will be very helpful.
  5. Please state the importance of using time scale for your figures 5 and 6 in the manuscript. Add this explanation to the manuscript.
  6. Section 3.4: Did you perform any quantification for the droplet rotation with respect to the properties of the core/shell?
  7. Please proofread the article for grammatical and spelling errors
